# ENTROPY-LENS: THE INFORMATION SIGNATURE OF TRANSFORMER COMPUTATIONS

## ABSTRACT

Transformer models map input token sequences to output token distributions, layer by layer. While most interpretability work focuses on internal latent representations, we study the evolution of these token-level distributions directly in vocabulary space. However, such distributions are high-dimensional and defined on an unordered support, making common descriptors like moments or cumulants ill-suited. We address this by computing the Shannon entropy of each intermediate predicted distribution, yielding one interpretable scalar per layer. The resulting sequence, the *entropy profile*, serves as a compact, information-theoretic signature of the model's computation. We introduce `Entropy-Lens`, a model-agnostic framework that extracts entropy profiles from frozen, off-the-shelf transformers. We show that these profiles (i) reveal family-specific computation patterns invariant under depth rescaling, (ii) are predictive of prompt type and task format, and (iii) correlate with output correctness. We further show that Rényi entropies yield similar results within a broad range of $\alpha$ values, justifying the use of Shannon entropy as a stable and principled summary. Our results hold across different transformers, requiring only forward access to intermediate hidden states and the output head; no gradients or fine-tuning are needed.

## 1 INTRODUCTION

Transformer-based architectures (Vaswani et al., 2023) operate by iteratively mapping token inputs to distributions over next-token outputs, layer by layer (Shan et al., 2024). This view emphasizes their inherently probabilistic nature: each layer produces a distribution over the vocabulary, encoding the model's beliefs at that stage.

Despite their success across domains—from language to biology to vision (Devlin, 2018; Ji et al., 2021; Wu et al., 2020)—the internal evolution of these distributions remains poorly understood, resulting in unpredictable behaviour (Wei et al., 2022) and reliability concerns (Schroeder & Wood-Doughty, 2025; Huang et al., 2025). Most interpretability research either focuses on toy models (Elhage et al., 2021; 2022) or requires training a set of probes (Belrose et al., 2023) or full models on ad-hoc tasks (Nanda et al., 2023), limiting scalability and generality. Moreover, existing methods primarily study latent representations—hidden states, attention maps, or MLP activations (Nanda & Bloom, 2022; Bereska & Gavves, 2024; Chefer et al., 2021)—rather than the output distributions themselves. We propose to shift perspective: instead of analyzing the internal geometry of latent activations, we focus on the evolution of the model's token-level predictions across layers. These predictions, obtained by decoding intermediate representations via the model's output head, form probability distributions over the vocabulary. Studying their evolution offers a direct, vocabulary-grounded view of the computation.

However, these distributions pose two challenges: they are high-dimensional (one value per token) and defined over an unordered support (the vocabulary). This makes classical descriptors like variance or higher-order cumulants unstable or ill-defined. We address both issues by computing the Shannon entropy of each distribution—a scalar, interpretable quantity invariant to token permutation, and reflective of the model's uncertainty at each layer. We introduce `Entropy-Lens`, a simple, scalable, and model-agnostic framework for studying transformer computations through the evolution of entropy across layers. Conceptually, `Entropy-Lens` can be seen as a dimensionality reduction of transformer activity, compressing complex layer-by-layer computations into an entropy profile that provides a compact and interpretable signature of the model's behavior.

In our experiments, we consider several LLMs, including Llama (Touvron et al., 2023), Gemma (Team et al., 2024), and GPT, (Radford, 2018) up to 9B parameters. For each generated token, we compute the entropy of its intermediate predictions, as described in Section 4, yelding one interpretable value per layer: this constitutes the *entropy profile*. We then aggregate these profiles across tokens and study whether they capture structural information about the internal LLM's computation (Figure 2a). In Section 5, we show that: (i) entropy profiles reveal family-specific signatures of transformer architectures, invariant under depth rescaling; (ii) they are predictive of prompt semantics and format; (iii) they correlate with output correctness; (iv) and results are stable across Rényi entropy variants, justifying our use of Shannon entropy as a principled default.

`Entropy-Lens` requires no gradients or fine-tuning, and only forward access to intermediate hidden states and the output head. It applies to frozen, off-the-shelf transformers of arbitrary size. While our experiments focus on language models, the methodology is general and opens new directions for vocabulary-grounded interpretability.

## 2 RELATED WORK

**Lenses in LLMs.** Mechanistic interpretability (Bereska & Gavves, 2024) aims to provide a precise description and prediction of transformer-based computations. Common tools in the field are *lenses*, which are a broad class of probes deployed in intermediate steps of the residual stream. For example, `logit-lens` (nostalgebraist, 2020) uses the model's decoder function to decode the intermediate activations in the vocabulary space. `tuned-lens` (Belrose et al., 2023) refines this technique by training a different affine probe at each layer, instead of only using the pretrained model's decoder function. Building on the `Transformer-Lens` library (Nanda & Bloom, 2022), we propose `Entropy-Lens`, which employs `logit-lens` to study and characterize LLMs' computations via their decoded version with information theory.

**Information theory in Transformers.** Information theory has been studied both in connection to the training phase of LLMs and their interpretability. For example, a collapse in attention entropy has been linked to training instabilities (Zhai et al., 2023) and matrix entropy was employed to evaluate "compression" in LLMs (Wei et al., 2024). Additionally, mutual information was used to study the effectiveness of the chain-of-thought mechanism (Ton et al., 2024). Our work, instead, shifts the focus on the vocabulary's natural domain. Through `Entropy-Lens`, we use information theory to study the evolution of entropy of the intermediate layers' decoded logits. A related study by Dombrowski & Corlouer (2024) uses information-theoretic measures to distinguish between truthful and deceptive LLM generations. Their analysis focuses on deception under explicit instruction, whereas our work aims at uncovering broader entropy-based signatures across model families, prompt types, and output correctness without requiring behavioral framing or fine-tuned prompting. A further line of research connects information theory to memorization in transformers. Brown et al. (2021) quantified memorization using Shannon mutual information between training data and trained models, while Morris et al. (2025) extended the analysis with Kolmogorov information theory at the instance level. Their studies focus on memorization at the final layer, whereas we estimate memorization across all layers—by analyzing sub-models of increasing depth—and uncover non-trivial, context-dependent patterns that deviate from the monotonic trend one might expect a priori (see Appendix D.1 for a derivation linking our entropy measure to memorization).

## 3 BACKGROUND

### 3.1 INFORMATION THEORY

The main information-theoretic quantity used in our study is *entropy*. Given a discrete[1] random variable $X$ with outcomes $x_i$ and probability mass function $p$, the *Shannon entropy* $H$ of $X$ is defined as

$$H(X) = -\sum_i p(x_i) \log p(x_i) = \mathbb{E}[-\log p(X)]. \tag{1}$$

---

[1]Although entropy can be naturally extended to the continuous case with probability *density* functions, we restrict ourselves to the discrete case as it is the most relevant to our study.

Shannon proved that this function is the only one—up to a scalar multiplication—that satisfies intuitive properties for measuring 'disorder' (Shannon, 1948). These include being maximal for a uniform distribution, minimal for the limit of a Kronecker delta function, and ensuring that $H(A, B) \leq H(A) + H(B)$ for every possible random variable $A$ and $B$. The same function already existed in continuous form in physics, where it linked the probabilistic formalism of statistical mechanics with the more phenomenological framework of thermodynamics, where the term 'entropy' was originally coined (Gibbs, 1902). In addition to Shannon entropy, we also consider its generalization known as *Rényi entropy*. Given a discrete random variable $X$ with probability mass function $p$, the Rényi entropy of order $\alpha > 0$, $\alpha \neq 1$, is defined as:

$$H_\alpha(X) = \frac{1}{1 - \alpha} \log \sum_i p(x_i)^\alpha. \tag{2}$$

This formulation reduces to Shannon entropy in the limit $\alpha \to 1$, and introduces a tunable parameter $\alpha$ that modulates the sensitivity of the entropy to the distribution's tail. Rényi entropy also subsumes many classical descriptors of discrete distributions without intrinsic ordering: with appropriate choices of $\alpha$, it recovers collision entropy ($\alpha = 2$), min-entropy ($\alpha \to \infty$), and max-entropy ($\alpha \to 0$), and it correlates with indices such as the Gini–Simpson index (Rényi, 1961; Jost, 2006). In general, lower values of $\alpha$ give more weight to rare events, while higher values emphasize the most probable outcomes.

Next, we study the entropy of vocabulary predictions—a quantity that is maximal when the prediction assigns equal probability to all tokens, minimal when it assigns zero probability to all but one token, and takes intermediate values when probability is distributed across multiple tokens, consistent with the previously mentioned properties. In our experiments, we explored a range of $\alpha$ values to understand how this parameter affects the informativeness of the resulting entropy profiles. We identified an *informative range* of $\alpha$ values—which includes the Shannon case $\alpha = 1$—where entropy profiles retain high discriminative power for classification tasks. Outside of this range, we observe that profiles tend to collapse: for very small $\alpha$, entropies become nearly maximal and lose contrast across layers and examples; for large $\alpha$, they become very small and overly sensitive to noise and local fluctuations. These findings support the use of Shannon entropy as a balanced and parameter-free choice, robust across a wide range of practical conditions.

## 3.2 THE TRANSFORMER

**Architecture.** The transformer (Vaswani et al., 2023) is a deep learning architecture widely applied in language modelling with LLMs (Brown et al., 2020) and computer vision (Dosovitskiy et al., 2021). Transformer computations happen through *transformer blocks* and *residual connections*, as exemplified in Figure 1b. While various design choices are possible, blocks are usually a composition of layer normalization (Zhang & Sennrich, 2019), attention, and multi layer perceptrons (MLPs), as shown in Figure 1a. Residual connections, instead, sum the output of the layer $i - 1$ to the output of the layer $i$.

Inside a single transformer block, the information flows both *horizontally* and *vertically*. The former, enabled by the attention mechanism, allows the token representations to interact with each other. In a language modelling task, for example, this is useful to identify which parts of the input sequence—the sentence prompt—should influence the next token prediction and quantify by how much. The latter vertical information flow allows the representation to evolve and encode different meanings or concepts. Usually, the dimension of the latent space is the same for each block in the transformer. The embedding spaces where these computations take place are generally called the *residual stream*.

**Computation schema.** LLMs are trained to predict the next token in a sentence. That is, given a sentence prompt $S$ with tokens $t_1, \ldots, t_N$, the transformer encodes each token with a linear encoder $E$. Throughout the residual stream, the representation $\mathbf{x}_N$ of the token $t_N$ evolves into the representation of the token $t_{N+1}$, which is then decoded back into token space via a linear decoder $D$, often set to $E^\top$, tying the embedding weights and the decoder. Finally, the logits—the output of $D$—are normalized with $\mathrm{softmax}$ to represent a probability distribution over the vocabulary. We summarize this operation with the function $W := \mathrm{softmax} \circ D$.

In formal terms, information processing can be expressed using the encoder, decoder, Transformer

block $f$, and residual connection

$$\mathbf{x}_j^0 = E(t_j), \quad \mathbf{x}_j^i = f^i(\mathbf{x}_{[1:N]}^{i-1}) + \mathbf{x}_j^{i-1}, \quad \mathbf{y}_j^i = W(\mathbf{x}_j^i). \tag{3}$$

where $j \in \{1, \ldots, N\}$ ranges over the number of tokens in the prompt and $i \in \{0, \ldots, L\}$ ranges over the number of layers. Hence, $\mathbf{x}_j^i$ represents the activations of token $t_j$ after layer $i$.

### 3.2.1 Instruct models

Training an LLM requires vast amounts of data and is generally divided into multiple phases. (1) **Pretraining**: The model is exposed to large datasets through self-supervised tasks, such as next-token prediction or similar variants. This phase helps the model learn a broad range of general knowledge. (2) **Fine-tuning**: This phase teaches the model to generate more useful and coherent responses. Two main strategies are used: *Chat*: The model is trained on structured conversations between a user and the model, with clearly defined roles. *Instruct*: The model learns from simple commands, without a predefined dialogue structure. (3) **RLHF** (optional): Some models undergo Reinforcement Learning from Human Feedback (RLHF) to further refine their responses based on human preferences.
For our experiments, we used off-the-shelf models. We also focused primarily on Instruct models (abbreviated with `it` in tables and figures) instead of Chat models for two reasons: 1. the Instruct strategy aligns better with our experimental setup 2. Instruct models are more flexible and often preferred for practical applications.

## 4 Method

`Entropy-Lens`'s pipeline comprises three steps and is described in Figure 1b. After introducing the notation and motivating the choice of entropy as a measure, in the following we describe the details of the framework which has been used for the experiments in the following sections.

**Notation.** We denote the input sentence comprising tokens $t_1, \ldots, t_N$ by $S = (t_i)_{i=1}^N$. Then, $\mathbf{x}_j^i$ denotes the activations of the token $t_j$ after block $i$ for $j \in \{1, \ldots, N\}$ and $i \in \{1, \ldots, L\}$. Since our analysis focuses on the outputs extracted from the intermediate layers of the transformer, it will be useful to distinguish between the *raw* logits and their *normalized* versions, i.e. the probability distributions. We define $W := \mathrm{softmax} \circ D$ and $\mathbf{y}_j^i := W(\mathbf{x}_j^i)$, the probability distribution over the vocabulary obtained from the activations of token $t_j$ after layer $i$.

**Why entropy?** Our goal is to characterize how a transformer model evolves its predictions across layers, remaining in the token space for interpretability. Each intermediate representation, once decoded and passed through $\mathrm{softmax}$, yields a probability distribution over the vocabulary. However, this distribution lives on a high-dimensional, unordered categorical support. Classical descriptors such as variance or higher-order cumulants rely on an implicit ordering of the support and thus become meaningless when applied to token distributions—shuffling token indices alters their value arbitrarily. Entropy, on the other hand, is invariant under permutations and captures a well-defined notion of uncertainty or informativeness regardless of vocabulary indexing. Rényi entropy, in particular, further recovers or correlates with many of the measures commonly used to describe distributions over unordered supports, making it a principled and unifying choice. It is therefore a natural and stable summary to characterize the evolution of token-level beliefs across transformer layers.

**Definitions.** The core of our methodology is to analyze the entropy of the generated tokens' intermediate representations $\mathbf{y}_j^i$. These vectors are probability distributions, as they are the output of a $\mathrm{softmax}$. To obtain a single quantity that summarizes the information they contain, we compute their entropy $H(\mathbf{y}_j^i)$. For one generated token, we can consider the entropy of all of its intermediate predictions $H(\mathbf{y}_j^i)$ for $i \in \{1, \ldots, L\}$. This leads us to the definition of entropy profile:

**Definition 1 (Entropy profile)** *Let $h_j^i = H(\mathbf{y}_j^i)$ be the entropy of the intermediate representation of token $t_j$ after block $i$ and residual connection. The entropy profile of the next generated token is defined as $\mathbf{h}_N = \bigoplus_i h_N^i$ where $\bigoplus$ denotes any aggregation function.*

In our experiments, we set $\bigoplus$ to be concatenation, so that $\mathbf{h}_N = (h_N^1, \ldots, h_N^L)^\top$, but other choices are possible. The extraction of entropy profiles is the step 1 of our pipeline. Then, we fix the number of tokens that the LLM is required to generate, $T$ and repeat the same procedure for each of them, leading us to the next definition:

**Definition 2 (Aggregated entropy profile)** *Let $\mathbf{h}_{N+t}$ be the entropy profiles according to Definition 1 for $t \in \{0, \ldots, T-1\}$, i.e. the entropy profile of each token generated sequentially by a transformer. The aggregated entropy profile of the next $T$ generated tokens is defined as $\mathbf{h}_{[N:T]} = \bigotimes_{t=0}^{T-1} \mathbf{h}_{N+t}$ where $\bigotimes$ denotes any aggregation function.*

Note that $\bigotimes$ in Definition 2 need not be the same as $\bigoplus$ in Definition 1. In our experiments, we set both of them to be concatenation, so that $\mathbf{h}_{[N:T]}$ is the matrix with $\mathbf{h}_{N+t}$ as columns, that is $(\mathbf{h}_{[N:T]})_t^i = \mathbf{h}_{N+t}^i$ for $i \in \{1, \ldots, L\}$ and $t \in \{0, \ldots, T-1\}$. The aggregation of entropy profiles is the step 2 of our framework.

The last step of our framework is classification, where we feed the aggregated entropy profile to a classifier $\mathcal{C}$ to determine whether it contains sufficient information to identify a particular 'entity'. In our experiments, we take $\mathcal{C}$ to be a k-NN classifier.

We also examine whether aggregated entropy profiles identify model family (Section 5.1), task type and format (Sections 5.2 and 5.5), and correct and wrong answers to multiple choice questions (Section 5.3) in LLMs.

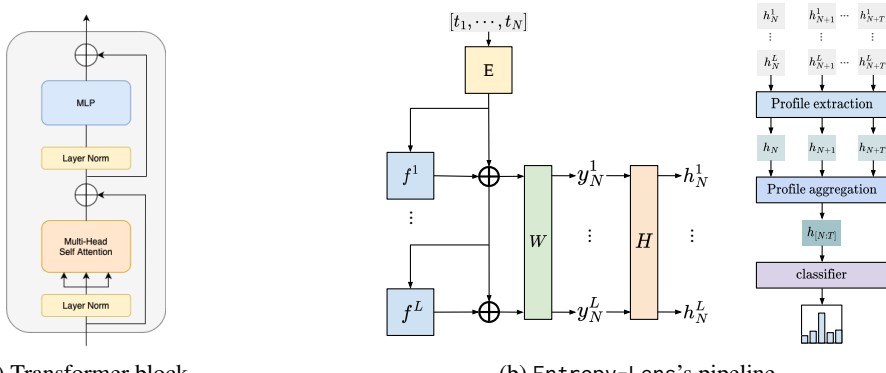

(a) Transformer block.  (b) Entropy-Lens's pipeline.

Figure 1: Overview of transformer computations and Entropy-Lens framework. (Left) Structure of a generic Transformer block. (Center) A diagram representing a transformer architecture: hidden representations are converted into intermediate predictions with $W$ before calculating their entropy $H$. (Right) A diagram representing our framework: step 1: entropy profile extraction, step 2: entropy profile aggregation and step 3: classification.

## 5 EXPERIMENTS

Our experiments focus on several key aspects. First, we show that entropy profiles are indicative of model family, with distinctions becoming more pronounced as model size increases. We then investigate whether a model's entropy profile alone can be used to classify the task it is performing. Next, we assess whether entropy profiles capture information about output formatting, independently of task content. We extend this analysis to evaluate whether entropy profiles provide signals that relate to correct versus incorrect task execution. Additionally, we explore the effect of varying the $\alpha$ parameter in Rényi entropy, examining how it influences the expressiveness of the resulting profiles. Notably, as entropy is a permutation invariant quantity, i.e. $H((p_1, \ldots, p_n)) = H((p_{\pi(1)}, \ldots, p_{\pi(n)}))$ for all permutations $\pi$, we do not have direct access to semantic information.

## 5.1 ENTROPY PROFILES IDENTIFY MODEL FAMILIES

We assess whether aggregated entropy profiles can distinguish different model families by visualizing and analyzing those of 12 models from 4 different families (GPT, Gemma, LLama and Qwen) with parameter counts ranging from 100M to 9B.

We average 64 entropy profiles obtained by generating 32 tokens prompting the model with a completely blank prompt. More details about the setup are available in Appendix C.1. We observe (see Figure 2a) that the profiles relate uniquely to the model family, rather than a particular model, independently of its size. Moreover, we observe that each model size within a particular family is tied to a scaling factor if we normalize by number of layers (see Figure 2b).

The GPT model class starts with high vocabulary entropy in the early layers, indicating a wide range of possible response tokens. Then, entropy gradually decreases—more smoothly than in other classes—leading to a low-entropy state, where the model narrows down to a small set of possible response tokens.

The Gemma model class, on the other hand, starts with high entropy in the very first layer, then sharply drops to lower entropy, rises again in the intermediate layers, and finally decreases to low entropy again just before the last layers, where the model is required to produce an output token.

The Llama model class starts with low entropy, then steeply rises and maintains a high entropy value over a large range of intermediate layers, finally decreasing to low entropy again.

The Qwen model family exhibits a similar trend, but in a more gradual manner, resulting in less well-defined regimes.

We observe that the equivalence between models of the same family but different sizes holds when looking at the entropy trend not as a function of the absolute layer index, but rather as the relative layer position within the model.

We conjecture that high entropy phases, whether in the early or intermediate layers, allow the model to explore more possibilities in its response, similarly to how temperature helps avoid getting stuck in local minima in optimization. Then, at the moment of selection, the distribution is 'cooled down', forcing the output to be limited to a few possible tokens.

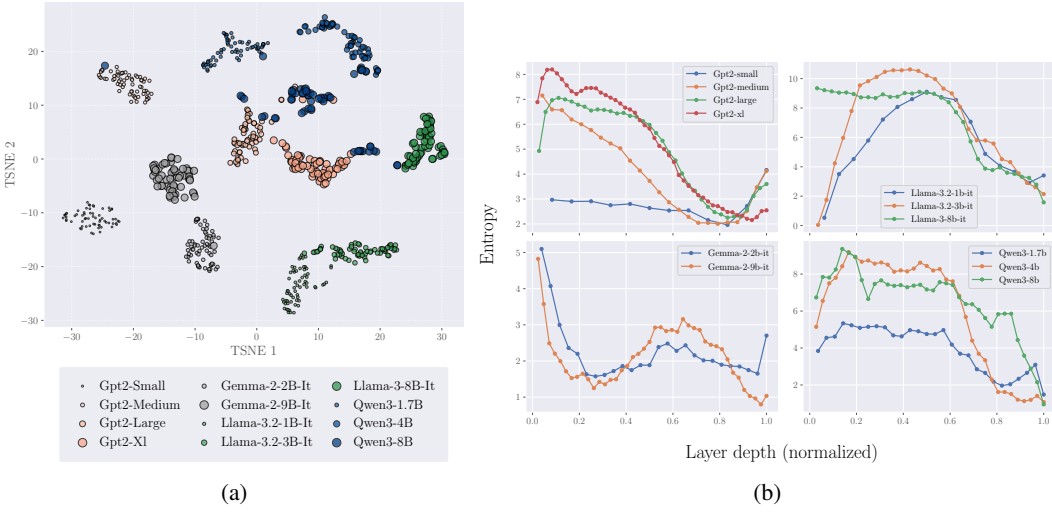

(a)  (b)

Figure 2: Entropy-based characterization of model families: (a) t-SNE of aggregated entropy profiles of different model families. Point size scales with number of parameters. (b) Average entropy profiles over 32 tokens per model. The $x$ axis is normalized to compare models with different depths.

## 5.2 ENTROPY PROFILES IDENTIFY TASK TYPES

We verify whether the entropy profiles can identify task types examining generative (continue a text), syntactic (count the number of words in a text), and semantic (extract the subject or moral of a text) tasks.

We do this with the *TinyStories* dataset (Eldan & Li, 2023). For evaluation robustness, we construct

for each task type three prompt templates using a combination of task-specific `task prompts`, reported in Appendix C.2 Table 4, and a `story` from *TinyStories*. These templates are: (1) **Base**, of the form `task prompt + story`; (2) **Reversed**, of the form `story + task prompt`; (3) **Scrambled**, of the form `task prompt + scrambled story` or `scrambled story + task prompt`, at random. A `scrambled story` is a 'story' obtained by randomly shuffling the words in a given `story` from *TinyStories*. Note that, for a robust evaluation, we also use 2 possible `task prompt` variations, as per Table 4.

We generate 800 prompts per task type, $1/3$ of them with the base template, $1/3$ with the reversed template, and $1/3$ with the scrambled templates, for a total of 2400 prompts. We then apply our pipeline from Section 4 to classify the aggregated entropy profiles of these prompts against their task type using a k-NN classifier. The model was evaluated in a 10-fold cross-validation using the ROC-AUC score (one-vs-rest), a standard choice for measuring classification performance. Table 1a shows the results obtained for 6 models with parameter counts ranging from 1B to 9B. Figure 3 shows the average entropy profiles per task type.

We observe that all k-NN classifiers (i.e. one for each LLM) achieve high accuracy in distinguishing entropy profiles, with a trend toward improved performance for larger models (see Appendix B.2 for a comparison with single-layer entropy baselines).

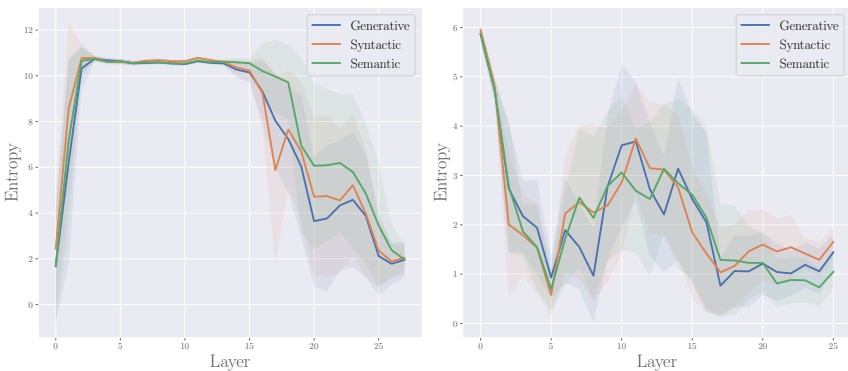

Figure 3: Average entropy profiles with shaded standard deviation for different task types: generative, syntactic, and semantic. These tasks are induced with the prompts described in Appendix C.2. Left: Llama-3.2-it. Right: Gemma-2-it.

Table 1: Results summary (ROC-AUC). Left: TinyStories task classification (Section 5.2). Middle: MMLU correctness vs. prompt style (Section 5.3). Right: Format classification vs. Rényi $\alpha$ (Section 5.5).

| (a) TinyStories | | | (b) MMLU correctness | | | | (c) Format vs. $\alpha$ | | |
|---|---|---|---|---|---|---|---|---|---|
| **Model** | **Size** | **k-NN AUC** | **Model** | **Prompt** | **LLM-Acc.** | **k-NN AUC** | **Model** | $\alpha$ | **k-NN AUC** |
| Gemma-2-it | 2.1B | $97.66 \pm 0.47$ | | Base | 50.89 | $73.61 \pm 1.52$ | | 0.5 | $97.3 \pm 1.6$ |
| Gemma-2-it | 8.9B | $98.38 \pm 0.50$ | Llama | Humble | 58.51 | $69.90 \pm 1.06$ | Gemma-2-2B-it | 1.0 | $98.7 \pm 1.1$ |
| Llama-3.2-it | 1B | $94.94 \pm 0.79$ | | Instruct | 60.62 | $67.23 \pm 1.62$ | | 5.0 | $98.4 \pm 1.7$ |
| Llama-3.2-it | 3B | $94.77 \pm 0.93$ | | Base | 56.10 | $71.88 \pm 1.63$ | | 0.5 | $97.8 \pm 1.6$ |
| Llama-3-it | 8B | $96.10 \pm 0.67$ | Gemma | Humble | 54.71 | $72.78 \pm 1.15$ | Llama-3.2-1B-it | 1.0 | $97.8 \pm 2.4$ |
| Phi-3 | 3.6B | $97.07 \pm 0.87$ | | Instruct | 56.38 | $68.36 \pm 1.23$ | | 5.0 | $96.6 \pm 2.6$ |

## 5.3 ENTROPY PROFILES CORRELATES WITH CORRECT TASK EXECUTION

We test whether entropy profiles can identify correct and wrong answers generated by LLMs using the Massive Multitask Language Understanding (MMLU) dataset (Hendrycks et al., 2021). MMLU consists of multiple-choice questions across 57 subjects, ranging from history and physics to law, mathematics, and medicine. The difficulty levels span from elementary to professional, making it a benchmark for evaluating language models on specialized domains. Each dataset entry contains: a question string, four answer choices and a label indicating the correct answer.

We evaluate two models, a Llama-3.2 with 3B parameters Instruct and a Gemma-2 with 2B parameters, by presenting the multiple-choice questions in three different formats (as per Table 5 in

Appendix C.3): (1) **Base**: A minimal version containing the topic, the question, and multiple-choice answers. (2) **Instruct**: A version with a brief explanation that it's a multiple-choice test where only one option should be selected. (3) **Humble**: A version that also instructs the model to pick a completely random option if it doesn't know the answer.

Then, we apply our pipeline to extract and aggregate the responses' entropy profiles and classify them against the correctness of the corresponding LLM-generated answer. We train a k-NN classifier for each LLM and validate it using 10-fold cross-validation. We also conduct a t-test to compare our classifier to a dummy model. This dummy model generates predictions randomly, sampled from a distribution that reflects the proportion of correct and incorrect answers produced by the LLM, ensuring robustness against class imbalance. The results reject the null hypothesis ($\alpha = 0.05$) with the k-NN achieving a ROC-AUC score between 67.23 and 73.61, depending on prompt type and model (see Table 1b).

We observe that the instruct and humble prompts improve Llama's average accuracy, while for Gemma, this is only true for the instruct prompt. Additionally, in Llama, the model's higher accuracy seems to be partially linked to greater difficulty in distinguishing correct from incorrect entropy profiles, though more rigorous analysis is needed to confirm this. In Gemma, however, this claim is harder to support.

## 5.4 THREE REGIMES OF RÉNYI ENTROPY

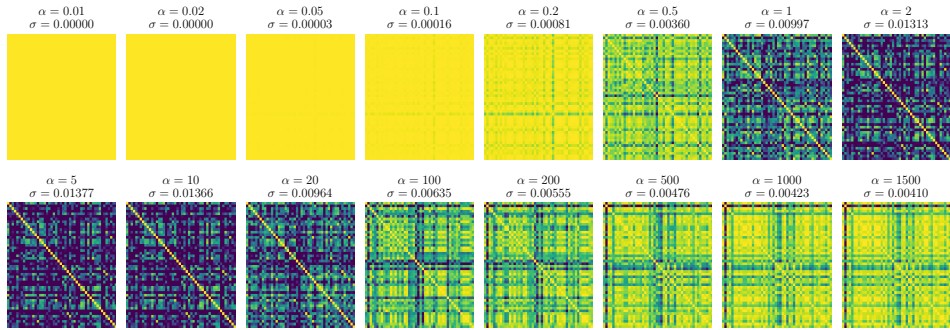

Figure 4: Cosine similarity matrices between entropy profiles computed on a subset of the *topic-format* dataset using different values of $\alpha$ in the Rényi entropy. $\sigma$ denotes the standard deviation of the similarity matrix. Note how similarity flattens for very low and very high $\alpha$, while intermediate values yield more informative profiles.

To qualitatively explore how the Rényi entropy affects the structure of entropy profiles, we compute pairwise cosine similarities between profiles generated with different values of $\alpha$. This analysis is performed on a subset of the *topic-format* dataset (see Appendix C.4), and the resulting similarity matrices are shown in Figure 4.

We observe three distinct regimes as $\alpha$ varies: (1) For very small values of $\alpha$ (e.g., $\alpha < 0.2$), the similarity matrices are nearly flat, with profiles being almost identical across examples. This is expected, as Rényi entropy in this regime weights all tokens with non-zero probability almost equally, and usually all tokens have non-null probability. (2) For large values of $\alpha$ (e.g., $\alpha > 20$), the similarity matrices also flatten. In this case, the entropy becomes increasingly dominated by the few tokens with highest probabilities. Since these sets of top tokens tend to have similar cardinalities (in the limit equal to 1), the profiles collapse into a narrow set of values, losing expressiveness and becoming more sensitive to local fluctuations. (3) Between these extremes lies an informative regime—approximately $0.5 \leq \alpha \leq 20$—where entropy profiles are heterogeneous enough to retain meaningful variation. This is reflected in the standard deviation of the similarity matrices, which peaks in this interval.

This qualitative observation supports our empirical findings: in Section 5.5, we show that format classification accuracy remains high and stable within this informative $\alpha$ range. Notably, Shannon entropy ($\alpha = 1$) falls within this interval, providing a strong justification for its use in the main experiments. By choosing $\alpha = 1$, we retain discriminative power while avoiding the need to tune additional hyperparameters.

## 5.5 ENTROPY PROFILES IDENTIFY TEXT FORMAT

We test whether the entropy profiles contain signal about the *format* of the generated text. In this experiment, we prompt the model to produce short texts across different topics while enforcing one of three predefined formats: `poem`, `scientific piece`, or `chat log`. We call these generated texts the *topic-format* dataset.

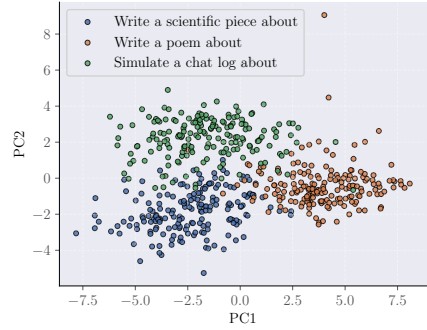

Figure 5: PCA projection on aggregated entropy profiles extracted from the *topic-format* dataset.

We use entropy profiles computed not only with Shannon entropy ($\alpha = 1$), but also with Rényi entropy for $\alpha \in \{0.5, 5.0\}$. As shown in Table 1c, the k-NN classifier achieves high ROC-AUC scores across all values of $\alpha$, indicating that the format is reliably encoded in the entropy profiles. Moreover, the small variability in performance across $\alpha$ values supports our use of Shannon entropy as a principled, parameter-free default.

To better understand the structure of the entropy profiles, we perform Principal Component Analysis (PCA) and visualize the first two components. As shown in Figure 5, the profiles cluster distinctly by format, forming linearly separable groups in the reduced space. This indicates that format-specific computation patterns are not only detectable by a classifier, but visibly reflected in the global shape of the entropy evolution across layers.

## 6 CONCLUSIONS

In this work, we prototyped a novel model-agnostic interpretability framework for large-scale transformer-based architectures grounded in information theory. `Entropy-Lens` can be interpreted as a dimensionality-reduction tool for transformer activity: it compresses complex computations into a simple profile that makes the model's behavior graspable at a glance. Across our experiments, we used `Entropy-Lens` to uncover family-specific computational patterns. Entropy profiles also proved informative of task type, format, and output correctness, and allowed us to identify which layers are more sensitive to these variations (Figure 3). Moreover, they were robust across Rényi entropy variants. Importantly, all experiments were conducted on frozen, off-the-shelf transformers without gradients or fine-tuning. From a more theoretical perspective, our analysis suggests that entropy profiles can be read in terms of memorization across depth (considering one layer, then two, and so on, as explained in Appendix D). Interestingly, our results indicate that this memorization is not monotonic, but instead depends systematically on family, task, and format—phenomena not previously observed. Finally, we emphasize that `Entropy-Lens` opens the door to further analyses, much like t-SNE or PCA do for representation spaces, which we leave to future work.

### 6.1 LIMITATIONS AND FUTURE WORK

While this work paves the way to further investigations in information theoretic interpretability, it also presents a number of limitations. First, the concepts of 'task type' and 'format type' don't have a formal and well established definition. Moreover, we showed how different models possess different characteristic entropy profiles. We conjecture that these particular shapes are a byproduct of training procedure and architectural designs, but future research could focus on understanding the precise connections. Another interesting line of research could focus on considering more fine-grained measures of information instead of just an aggregated one such as entropy. With these limitations in mind, our methodology could be used to probe the reasoning capabilities of LLMs, for instance by comparing the entropy profile of a reasoning task vs. a data retrieval task. If these profiles happen to match, it could be taken as an argument against the ability of LLMs to reason. Conversely, if they do not, it may suggest that some form of reasoning is indeed occurring.

Finally, recent literature explored the use of entropy for private inference (PI), where computations are performed on encrypted data without revealing users' sensitive information (Jha & Reagen, 2025). While previous work focused on the entropy of the attention mechanism, future research could use our methodology to develop PI-friendly applications of LLMs.

## REPRODUCIBILITY STATEMENT

To ensure reproducibility of our results, all code necessary to reproduce the experiments presented in this paper is available in the source code included in the supplementary materials. Complete details about the hardware specifications and software libraries used are provided in Appendix C.5.

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

# ENTROPY-LENS: THE INFORMATION SIGNATURE OF TRANSFORMER COMPUTATIONS - APPENDIX

## A APPENDIX STRUCTURE

The appendix is organized as follows:

- **Appendix B – Additional Experiments:** We provide a preliminary exploration of our approach on Vision Transformers (ViTs), showing that entropy profiles can also be extracted and qualitatively interpreted in non-language domains, without any modification to our framework. We also conducted a supplementary experiment on the TinyStories dataset (using the Gemma-2-2b-it model) to determine which transformer blocks are essential for the classification task and whether all blocks are necessary for optimal performance.

- **Appendix C – Evaluation Details:** We provide full details on the datasets and prompt templates used in our experiments. In particular, we highlight details about the model family identification, we specify how prompts were constructed for task classification, output correctness, and format classification tasks. This section also includes hardware information to support reproducibility

- **Appendix D – Theoretical Considerations:** We outline a heuristic connection between entropy profiles and memorization in transformers. Building on the frameworks of Brown et al. (2021) and Morris et al. (2025), we show how our layer-wise entropy measures can be interpreted as estimates of memorization across depth.

- **Appendix E – Minimal Implementation:** We present a minimal code snippet that reproduces the core entropy profile extraction logic in a few lines of code. While our full codebase offers several optimizations and utilities, this section emphasizes transparency and ease of replication by showcasing the conceptual simplicity of our approach.

## B ADDITIONAL EXPERIMENTS

To further test the generality and flexibility of our methodology, we conduct additional experiments beyond the core settings presented in the main text. In particular, we explore how `Entropy-Lens` performs in a different modality: computer vision. Without any architectural adjustment or fine-tuning, we apply the same framework to Vision Transformers (ViTs) and observe that entropy profiles extracted from visual models exhibit qualitatively interpretable structure. These preliminary results suggest that our method may extend beyond language models, but a systematic evaluation across modalities is left for future work.

We further performed a complementary study on the TinyStories dataset, using the Gemma-2-2b-it model, to assess which transformer blocks are most critical for task classification and whether all layers are necessary. In this setting, we compared full entropy profiles with reduced variants obtained from single layers or from equidistant subsets of layers (first, middle, and last). Our results show that the complete entropy profile achieves substantially higher classification accuracy, indicating that information is distributed across depth and cannot be captured by a small subset of layers alone.

### B.1 ENTROPY-LENS FOR VISION TRANSFORMERS

To demonstrate the versatility and robustness of our approach beyond language modeling, we analyze the entropy profiles of ViTs and DeiTs.

Using 20 classes from ImageNet-1K (Russakovsky et al., 2015), with 20 images per class, and without any modifications to our framework, we generate the entropy profiles shown in Figure 6. We observe that all profiles start with high entropy values, which then decrease, mostly in the final layers. This behavior is qualitatively similar to that of GPTs or larger LLaMa models (Section 5.1), pointing to a possible common trend across domains as different as image processing and natural language processing.

Focusing on computer vision models, we note that while ViT and DeiT families exhibit qualitatively similar trends, they differ quantitatively—ViTs start with higher entropy values, making them easily

distinguishable from DeiTs.

Notably, the only profile that stands out is that of ViT Large (with $\sim 300M$ parameters), compared to the other models analyzed in this section, which have $\leq 86M$ parameters.

For ViT Large, entropy decreases more smoothly, appearing not only as a better approximation of the sharp drop seen in smaller models but possibly following a different behavior entirely, with the entropy decline starting earlier.

We hypothesize a phase transition in entropy behavior as model size increases, occurring somewhere between 87M and 307M parameters, though a more extensive study would be required to confirm this hypothesis.

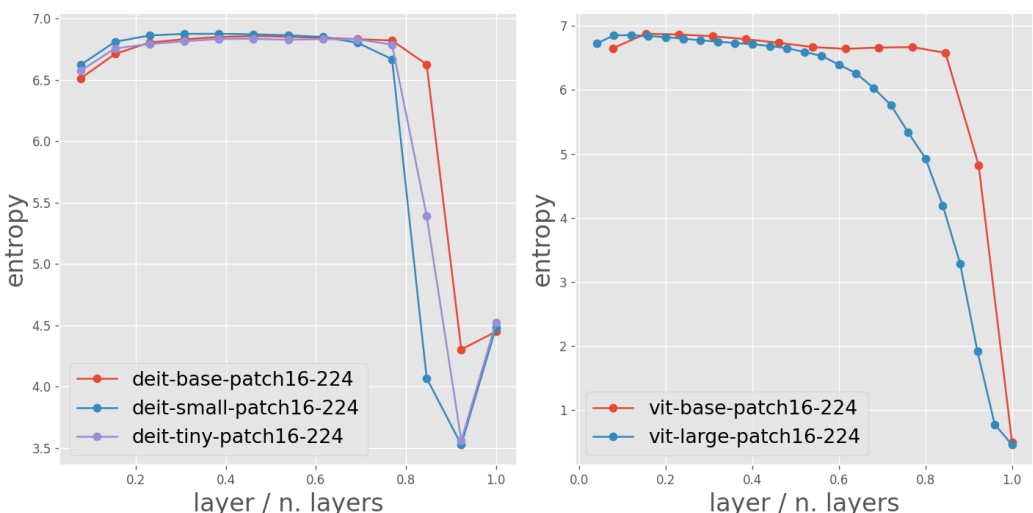

Figure 6: Entropy profiles for ViT model families.

## B.2 ENTROPY PROFILES FROM DIFFERENT BLOCKS

To assess whether entropy profiles from all transformer layers are necessary for model characterization, or if comparable results can be achieved using fewer layers, we conducted an evaluation using different layer subsets. Specifically, we repeated the *TinyStories* experiments (Section 5.2) using four different configurations: (1) first layer only, (2) middle layer only, (3) last layer only, and (4) a combination of first, middle, and last layers. We then compared the classification accuracy of k-NN classifiers trained on these reduced entropy profiles against those using complete layer sequences. The results are visible in table 2.

**Experimental setup.** The k-NN classifier was configured with $k = 11$ neighbors using Euclidean distance as the similarity metric. For sequence generation, we employed a sampling-based approach rather than deterministic decoding which was used for the results reported in the Section 5.2.

Table 2: k-NN AUC across different sets of considered layers for Gemma-2-2b-it.

| Considered layers | k-NN AUC |
|---|---|
| first-only | 68.34±2.68 |
| middle-only | 78.83±3.07 |
| last-only | 76.78±2.36 |
| first+middle+last | 86.13±1.41 |
| all | 90.49±1.76 |

# C EVALUATION DETAILS

This section provides further details on the datasets and prompt templates used to evaluate the effectiveness of entropy profiles in the main experiments. In particular, we describe how we constructed the inputs for three key experimental settings: task type classification using the *TinyStories* dataset, correctness classification using the MMLU benchmark, and format classification using the *topic-format* dataset.

In all cases, prompt design plays a critical role in ensuring robust comparisons across experimental conditions. To this end, we employed multiple prompt variations. The subsections below report the full set of templates used, as referenced in Sections 5.2, 5.3, and 5.5 of the main paper.

The scripts used to generate these datasets—along with the full codebase to extract entropy profiles and reproduce all experiments—are shared as part of our code release.

Finally, we also provide information about the hardware used to run our experiments to facilitate reproducibility.

## C.1 MODEL FAMILY CLASSIFICATION

In Section 5.1, we show how entropy profiles can effectively identify both model families and model sizes. Our analysis reveals that entropy profiles exhibit qualitatively distinct patterns across different model families and sizes, as illustrated through scatterplot visualizations. By applying t-SNE dimensionality reduction, we cluster models by family, indicating that entropy profiles capture meaningful structural differences between architectures. To quantitatively assess the classification capabilities of entropy profiles, we employ a k-nearest neighbors classifier ($k = 3$ and euclidean distance) to predict both model families and sizes based on their entropy traces. The classification results are presented in Table 3. To obtain labeled model size categories, we binned models into 4 classes based on parameter count (in billions): $<$1B, 1-3B, 3-5B, and $>$5B.

Table 3: F1-scores for model family and model size classification. Each reported value is the mean across 10 runs, with the standard deviation computed over random 50/50 train–test splits.

| Task | Macro F1-score |
|---|---|
| model family | 97.99±0.66 |
| model size | 96.31±0.87 |

**Preprocessing Steps.** Since entropy traces vary in length across models due to different layer counts, we apply linear interpolation to standardize all traces to the same length. Additionally, to ensure fair classification performance, we standardize the samples to reduce bias from scaling effects in the entropy profiles, allowing the classifier to focus on the characteristics of each trace.

## C.2 PROMPT TEMPLATES FOR TINYSTORIES TASKS

In Section 5.2, we describe an experimental setup designed to test whether entropy profiles can identify different types of tasks. To this end, we used the *TinyStories* dataset (Eldan & Li, 2023) and constructed prompts combining short stories with specific task instructions. Each task type—*generative*, *syntactic*, and *semantic*—was associated with two distinct natural language formulations, referred to as `task prompts`. These are listed in Table 4. By varying the task prompt, we ensure that our classification results are not simply driven by surface-level textual artifacts, but instead reflect deeper computational signatures captured by the entropy profiles.

This table complements the prompt templates (base, reversed, and scrambled) described in the main text and provides the full set of instructions used to elicit different model behaviors.

## C.3 PROMPT TEMPLATES FOR MMLU CORRECTNESS CLASSIFICATION

In Section 5.3, we test whether entropy profiles can distinguish between correct and incorrect answers produced by language models. To construct the dataset for this experiment, we used the

Table 4: Prompt templates used for *TinyStories* tasks.

| Task Type | Task prompt |
|---|---|
| Generative | How can the story be continued?
Which could be a continuation of the story? |
| Syntactic | How many words are in the story?
Count the number of words in the story. |
| Semantic | What is the main idea of the story?
Who is the subject of the story? |

Massive Multitask Language Understanding (MMLU) benchmark (Hendrycks et al., 2021), applying three distinct prompt styles to elicit different answer behaviors from the models. Table 5 reports the full prompt templates used in this experiment. Each template presents the same multiple-choice question in a different instructional format: the **Base** format presents the question directly; the **Instruct** format introduces an explicit instruction to select a single correct answer; and the **Humble** format includes an additional fallback directive encouraging the model to guess randomly if uncertain.

This variation in prompting allows us to control for instruction framing and to evaluate whether entropy profiles can capture response confidence and correctness robustly across different model behaviors. The table shown here complements the description in the main text.

Table 5: Prompt templates used for the MMLU dataset.

| Prompt Type | Prompt |
|---|---|
| Base | Subject: {subject}
Question: {question}

Choices:
A. {option_1}
B. {option_2}
C. {option_3}
D. {option_4}

Answer: |
| Instruct | The following is a multiple-choice question about {subject}. Reply only with the correct option.

Question: {question}

Choices:
A. {option_1}
B. {option_2}
C. {option_3}
D. {option_4}

Answer: |
| Humble | The following is a multiple-choice question about {subject}. Reply only with the correct option.
If you are unsure about the answer, reply with a completely random option.

Question: {question}

Choices:
A. {option_1}
B. {option_2}
C. {option_3}
D. {option_4}

Answer: |

### C.4 Prompt Construction for the *Topic-Format* Dataset

To evaluate whether entropy profiles captures stylistic features of generated text, we constructed a custom dataset referred to as the *topic-format* dataset. In this setting, models are prompted to generate short texts on various topics, each constrained to adopt one of three specific formats: poem, scientific piece, or chat log. The goal is to determine whether these formats induce distinct entropy profiles, independently of the topic content.

We generated prompts by pairing 150 distinct topics with the following three format instructions:

- Write a poem about ...
- Write a scientific piece about ...
- Simulate a chat log about ...

Each prompt is constructed by concatenating a format prefix with a randomly selected topic (e.g., Write a poem about a planet). The resulting dataset contains 450 prompt completions per model, each paired with its entropy profile computed using Rényi entropy for $\alpha \in \{0.5, 1.0, 5.0\}$.

All generations were performed using a maximum generation length of 256 tokens. We then segmented the output into 8 equal-length windows and computed an entropy profile for each. The resulting data were stored with format labels and used in the classification and visualization tasks discussed in Section 5.5 of the main paper.

This setup enables robust testing of the extent to which entropy profiles encode formatting cues, beyond topical content or task semantics.

### C.5 Experimental and hardware setup

All experiments were conducted on a compute node equipped with an NVIDIA L40 GPU, an Intel Xeon Gold CPU, 128 GB of RAM, and running Ubuntu 22.04. The primary software frameworks used were PyTorch, Transformer-Lens, and HuggingFace Transformers. Inference on LLMs was performed using float16 precision for improved efficiency.

## D  Theoretical Considerations

In addition to the empirical results presented in the main text, we provide here a preliminary theoretical perspective that connects entropy profiles to the literature on memorization in transformers. Our goal is not to give a full formal treatment, but rather to outline how previous definitions of memorization, based on information theory, can be related to the quantities we compute. We first recall the frameworks introduced by Brown et al. (2021) and Morris et al. (2025), and then show how our entropy profiles can be interpreted as layer-wise estimates of memorization.

### D.1 Entropy and Memorization

**From Shannon to Kolmogorov memorization.**   Brown et al. (2021) introduced an information-theoretic framework to quantify memorization in trained models. Given a training data distribution $X$, a family of data-generating processes $\Theta$, and a training algorithm $L : X \mapsto \hat{\Theta}$ mapping training sets to trained models, they define memorization as the mutual information between $X$ and $\hat{\Theta}$

$$\text{mem}(X, \hat{\Theta}) = I(X, \hat{\Theta}) = H(X) - H(X|\hat{\Theta}). \tag{4}$$

This quantity captures how much information about $X$ is retained in the distribution over trained models. It can be decomposed into

$$\text{mem}(X, \hat{\Theta}) = \text{mem}_I(X, \hat{\Theta}, \Theta) + \text{mem}_U(X, \hat{\Theta}, \Theta), \tag{5}$$

where $\text{mem}_I$ measures generalization and $\text{mem}_U$ the unintended memorization (i.e., information about $X$ not attributable to the process $\Theta$).

Building on this formalism, Morris et al. (2025) extended the analysis from distributions to individual instances, moving from Shannon to Kolmogorov information theory. The Kolmogorov

complexity of an instance $x$ given model parameters $\hat{\theta}$ is

$$H^k(x|\hat{\theta}) = \min_s \{|s| : f(s, \hat{\theta}) = x\}, \tag{6}$$

where $f$ is a computational model (imagine a decoder) that can take as input $x$ and $\theta$. The exact definition of Kolmogorov complexity is not computable in general. In practice, it can be approximated via arithmetic coding as

$$H^k(x|\hat{\theta}) \approx -\log p(x|\hat{\theta}), \tag{7}$$

where $p(x|\hat{\theta})$ is the predictive probability assigned to $x$ by the trained model. This allowed Morris et al. (2025) to study instance-level memorization, although still focusing on measures computed at the final layer of the model.

**Connecting to entropy profiles.** Our approach provides a complementary perspective. Instead of measuring memorization only at the final layer, we estimate it at every layer when we analyze entropy profiles. To see this connection, recall that in Morris et al. (2025) the term $H^k(x|\hat{\theta})$ is approximated by $-\log p(x|\hat{\theta})$, where $p(x|\hat{\theta})$ is the model's predictive distribution for instance $x$. In our notation, this probability corresponds to a component of the vector $\mathbf{y}_j^i$, the softmax-normalized output obtained for token $t_j$ after block $i$. Averaging this quantity with respect to the distribution $p(x|\hat{\theta})$ yields an estimate of $H(X|\hat{\theta})$. If we further assume that the distribution $\hat{\Theta}$ induced by the training algorithm is sufficiently concentrated around the trained model, this becomes close to $H(X|\hat{\Theta})$, the conditional entropy of the data given the trained model distribution.

Now, rather than computing this value only for the full model, we do so for every intermediate truncation. Let $\hat{\Theta}_i$ denote the sub-model obtained by retaining only the first $i$ layers of the trained transformer and applying an early exit. The corresponding conditional entropies are

$$H(X|\hat{\Theta}_i), \quad i = 1, \dots, N, \tag{8}$$

and the sequence $\{H(X|\hat{\Theta}_i)\}_{i=1}^N$ constitutes the *entropy profile*. This is equivalent to

$$H(X|\hat{\Theta}_i) = H(X) - I(X, \hat{\Theta}_i), \tag{9}$$

i.e. the negative mutual information between the dataset and the truncated model up to a constant $H(X)$, which is equal for all layers.

This perspective suggests that entropy profiles can be interpreted as measuring how memorization is distributed across depth. Crucially, our empirical results show that this allocation does not follow a simple monotonic trend, as one might have expected a priori. Instead, it varies in a systematic way depending on model family, task, format, and confidence, revealing non-trivial patterns of information storage that had not been documented before.

# E  MINIMAL IMPLEMENTATION

To maximize reproducibility and transparency, we provide a minimal implementation of our framework. While our full codebase includes optimizations and utility functions to streamline analysis across models and datasets, the core idea behind `Entropy-Lens` is conceptually simple and can be expressed in just a few lines of code.

This section presents a compact example that computes the entropy profile of generated tokens using an off-the-shelf language model. Despite its brevity, this snippet captures the essence of our method: extracting intermediate representations, mapping them to vocabulary distributions, and computing their entropies.

## E.1  MINIMAL ENTROPY PROFILE EXTRACTION

The code in listing 1 demonstrates how to compute an entropy profile for a single prompt using a standard decoder-only transformer. It relies only on model forward passes and the use of `logit-lens`-style decoding. No gradients or fine-tuning are required, but only forward access to intermediate hidden states and the output head.

```python
from transformers import AutoTokenizer, AutoModelForCausalLM
import torch

# Load GPT-2 and set up
tokenizer = AutoTokenizer.from_pretrained('gpt2')
model = AutoModelForCausalLM.from_pretrained('gpt2', device_map="auto").eval()
tokenizer.pad_token = tokenizer.eos_token

# Define entropy computation
ln, U = model.transformer.ln_f, model.lm_head
entropy = lambda x: -torch.sum(x * torch.log(x + 1e-15), dim=-1)

# Prepare input
input_text = 'The concept of entropy'
inputs = tokenizer.encode(input_text, return_tensors="pt").to(model.device)

# Generate with hidden states
outputs = model.generate(inputs,
    do_sample=True,
    max_new_tokens=32,
    output_hidden_states=True,
    return_dict_in_generate=True,
    pad_token_id=tokenizer.pad_token_id
)

# Stack hidden activations and compute entropy signature
activations = torch.vstack([
    torch.vstack(h).permute(1, 0, 2) for h in outputs.hidden_states
])
entropy_signature = entropy(U(ln(activations)).softmax(dim=-1))
```

Listing 1: A minimal Python implementation of Entropy-Lens using Huggingface models and Pytorch.

