# OpenReview forum: "Entropy-Lens: The Information Signature of Transformer Computations"
_ICLR.cc/2026/Conference — Submitted to ICLR 2026_

### Official Review · Reviewer_r1uB · 2025-10-24

**Soundness:** 4
**Presentation:** 4
**Contribution:** 2
**Rating:** 4
**Confidence:** 4

**Summary:**

The authors introduce the entropy of the logit-lens distribution at each layer as a measure. The goal is to understand how information is compressed over the depth of learning. The authors use this to establish that LLMs of different architectures are processing information differently. In particular, they compare and classify the entropy curves (over layers) for different architectures and model sizes, finding that similar models have similar global information processing properties. In addition, they show that different text types exhibit different profiles.

**Strengths:**

- The authors communicate their ideas clearly and precisely
- The entropy is indeed an interesting measure of the way information is processed in the network, and as the next token distribution is also favouring a probabilistic view, a natural one indeed.
- Several larger trained open source models are compared.

**Weaknesses:**

- While idea of using the entropy in interpretability intuitively seems useful, the experiments and discussion lack a grounded interpretation of the value and curve shapes themselves. This would be highly useful to understand whether the measure is practical and interpretable, or whether it only reflects statistical properties of the architecture or text inputs, which unavoidably lead to discriminatory entropy curves. To me, a successful *interpretable* analysis would connect instance-based observations and phenomena to the type of the curve, and validate that the entropy curve has some predictive properties. This deeper understanding is what I would have expected e.g. in the discussion of Section 5.2, and more generally in the experiments.

**Questions:**

Comments
- In Fig.2 b), it could also be useful to see some standard deviations.
- As far as I understand, your method aggregates entropy over generated sentences, hence the temperature in the generation might also play a role in your measure. How does this influence your results?
- An ablation on the role of the different architectures could be comparing the entropy profiles of trained and non-trained models. At least that way one might understand if the entropy of learned models is smaller (?) to than that of the untrained ones, and which of its properties come from being learned.

---

> ### Author Response · Authors · 2025-12-03
>
> We thank the reviewer for acknowledging the clarity of our presentation and the comprehensiveness of our experimental suite (excellent scores in both fields). We are grateful that the Reviewer found value in our comparisons across large open-source models and in our methodological framing. We address the reviewer’s concerns in the following.
>
> **Comment 1**
> Thank you for the suggestion. We agree that showing variability would clarify the robustness of the trend, and we will include standard deviations in the next revision.
>
> **Comment 2**
> Preliminary experiments indicate that the sampling temperature has only a minor effect on the aggregated entropy values. Nonetheless, a more systematic evaluation would be valuable, and we plan to include a dedicated analysis in future iterations of the paper.
>
> **Comment 3**
> We have started investigating this direction. Early results suggest that the entropy profile is almost uniform at initialization and rapidly develops its characteristic shape during training. This behaviour appears consistent across datasets, and more associated with architectural choices.
>
> We thank the Reviewer again for the constructive feedback, the valuable suggestions for further analysis, and the strong positive assessment of our paper’s soundness and presentation. We believe the Reviewer’s feedback will further strengthen the contribution.

---

### Official Review · Reviewer_PJ6w · 2025-10-26

**Soundness:** 3
**Presentation:** 2
**Contribution:** 2
**Rating:** 6
**Confidence:** 4

**Summary:**

The main contribution of the paper is simple and interesting --- an "entropy-lens" framework to compare models by considering the entropy of the output distribution induced at the head of each layer, for each input token.  They use this framework to identify patterns corresponding to model families, task types (generative/syntactic/semantic), and output correctness.

**Strengths:**

1. The paper studies a relevant and trending problem of trying to understand and analyze the inner workings of transformer models --- it does so by proposing a framework to "see inside the black-box".
2. The central idea (using the "entropy profile" to identify attributes of models/use-cases) is novel and well-motivated.
3.  The examples chosen to illustrate the utility of the "entropy-lens" framework are relevant and thought-provoking.

**Weaknesses:**

My major criticism of the paper is that though it provides an interesting framework to compare models, it is unclear what the higher goal here is, or how it fits towards the broader goal of understanding the inner workings of transformers.  They identify and observe interesting patterns (through their own novel method), but then offer little to no insights as to how/why these patterns show up.

For instance, they observe that different model families show consistently different trends in the entropy profile across layers --- but why does this happen?  does it affect model performance (concretely)?  does it suggest that some models are better suited to specific tasks?  I would have liked to see more answers here, possibly even using some smaller toy models to study these characteristics, it would have made the paper more insightful, in my view.  The same comment holds for the other observations as well --- the lack of a serious (i.e. supported by experiments, analysis) attempt at an explanation means that the merits of the paper are only to propose the entropy-profile and apply it (almost superficially) to a handful of (interesting) tasks.

As such, my score is currently to accept, but I would have liked some insights to support the paper strongly.  I will be happy to raise my score if the paper is able to add content along these lines, at least for one of the settings considered.

**Questions:**

1. "classical descriptors such as moments or cumulants unsuitable" is repeated several times, but it is still unclear to me why this is the case.  Do you have a clear explanation, possibly with references, for why this is so?  Yes, the output distributions are high-dimensional, but why is a moment "unsuitable" (as the paper claims, without specifying how) while an entropy is fine?

2. Have you considered any other choices of aggregators?  I would have thought that there might be patterns in the entropy profiles observed at different tokens within the same layer that might allow for a more efficient representation than having the entropies at ALL tokens.  It would also be interesting to see if this somehow connects to the problem of prompt compression (where the goal is to remove tokens that have low influence (ergo entropy) in the prompt).

3. For the "task type" experiments in Section 5.2, the curves in each plot of Fig. 3 look rather similar.  Is the paper suggesting that k-NN gets an AUC ~>95% in distinguishing among these curves?  The differences in the curves are curiously specific (marked by jumps/drops at specific positions)...

4. Section 5.5:  to really support the claim that Shannon entropy lies in the "informative" range of alpha values, it would be nice to have seen the k-NN AUC in Table 1(c) degrade for alpha outside the (0.5, 5) interval.  Does this happen?

Minor comments:
1. would be useful to have some references directing to how Renyi entropy correlates with other measures
2. Fig. 1(a):  is this one or two transformer blocks?  does each block not require all of attention, MLP, layer norm (as suggested in the text of Section 3.2)?
3. right after (3), it is not immediately clear what $f^i$ is, it would be useful to write "...where $f^i$ is the transformer block at layer $i$..."

---

> ### Author Response · Authors · 2025-12-03
>
> We thank the Reviewer for the detailed, constructive, and overall positive assessment. We especially appreciate the acknowledgment of the novelty and relevance of the entropy-lens framework. Below we address all points in turn.
>
> **Question 1**
> The central issue is that the support of the output distribution lacks any canonical ordering. Because moments and cumulants depend on the numerical ordering of support points, different arbitrary permutations lead to different values. For example, consider a four-point support where most mass lies on elements 1 and 4 under some ordering S\_1​; permuting elements to obtain a different valid ordering S\_2 alters the variance despite the underlying categorical distribution being identical. Since there is no principled way to choose among such orderings, these descriptors are not well-defined for our setting. In contrast, entropy is invariant to permutations of the support and therefore provides a stable and order-independent summary.
>
> **Question 2**
> We agree that selecting subsets of tokens or using more compressed representations may be appealing. However, doing so requires defining a selection criterion, and this risks introducing unintended biases that confound comparisons across models or layers. Our current choice to retain all tokens avoids such confounding effects. Exploring principled token selection strategies—ideally ones that preserve invariance properties—would indeed be a valuable direction for future work.
>
> **Question 3**
> The curves in Fig. 3 display mean profiles aggregated across runs, while the k-NN classifier operates on the full, unaveraged profiles. We are currently examining which specific layers drive the discriminative power. Nonetheless, the fact that these differences remain detectable by a simple classifier highlights that the entropy profiles retain task-specific structure and illustrates one concrete application of the proposed metric.
>
> **Question 4**
> This is a valuable suggestion and an analysis we intend to expand. Qualitatively, for very small α\\alphaα, Rényi entropy becomes dominated by the full support, yielding profiles that closely resemble those from Shannon entropy and thus reduce discriminability. For very large α\\alphaα, the measure focuses almost entirely on the most probable tokens, making comparisons noisy and substantially weakening the signal. This behaviour is reflected in Section 5.4, offering both theoretical and empirical support for the existence of an informative intermediate range. We agree that a more systematic study would further strengthen the claim.
>
> **Minor Comment 1**
>
> \[1\] and \[2\], cited in the main paper, show how Rényi entropy subsumes many classical descriptors of discrete distributions without intrinsic ordering: for specific values of \\alpha, it recovers collision entropy (\\alpha=2), min-entropy (\\alpha \\to \\infty), and max-entropy (\\alpha \\to 0\) and it correlates with diversity indices such as the Gini–Simpson index.
>
> \[1\] Rényi, Alfréd. "On measures of entropy and information." *Proceedings of the fourth Berkeley symposium on mathematical statistics and probability, volume 1: contributions to the theory of statistics*. Vol. 4\. University of California Press, 1961\.
>
> \[2\] Jost, Lou. "Entropy and diversity." *Oikos* 113.2 (2006): 363-375.
>
> **Minor Comment 2**
>
> Fig. 1(a) depicts **one** transformer block.
>
> **Minor Comment 3**
>
> Thank you for the suggestion. We will make sure it’s more clear.
>
> We thank the Reviewer again for the constructive feedback. We appreciate the positive assessment and the opportunity to further strengthen the paper.

---

### Official Review · Reviewer_rW4J · 2025-10-31

**Soundness:** 3
**Presentation:** 4
**Contribution:** 2
**Rating:** 4
**Confidence:** 4

**Summary:**

The paper introduces the idea of entropy lens which can be used to compute (Shannon or Rényi) entropy of the distribution over vocabulary for a particular token and a particular layer of a transformer language model. There entropies can be aggregated into an entropy profile associated with a particular output text. The paper investigates how distinctive such profiles are for particular models and model families, and for specific prompt types, and also whether they correlate with the answer to a MC questions being correct. The results show that in general the profiles are fairly informative for the tested cases.

**Strengths:**

- The approach is straightforward and transparent, and grounded at a basic level in information theory
- The presentation is clear and very easy to follow
- The findings are not particulary suprising but intriguing

**Weaknesses:**

While the results are convincing and the methodology solid, what is less developed in the paper are the implications and/or applications.
Connection to memorization across model layers is mentioned in passing in the main paper and briefly explained in the appendix. A couple of other other use cases are mentioned but not developed further.

So while this is a clean and curious study, the significance is less clear. Combined with the lack of major methodological developments, the contribution of this paper is rather minor.

**Questions:**

The paper is clear, I did not have any points to clarify.

---

> ### Author Response · Authors · 2025-12-03
>
> We thank the reviewer for acknowledging the transparency and presentation of our methodology and for their interests in our results. While it is true that similar metrics have been used in prior work, our aim here is to examine more carefully what this particular metric actually captures and how far its interpretive power extends. We believe that clarifying its behaviour is valuable both for understanding existing applications (such as memorization across layers) and for informing future ones. We agree that further development of these implications would strengthen the paper.

---

### Official Review · Reviewer_CJe4 · 2025-11-05

**Soundness:** 2
**Presentation:** 4
**Contribution:** 2
**Rating:** 2
**Confidence:** 3

**Summary:**

In this paper, the authors introduce entropy as a measure to study intermediate representations in transformers. By creating an entropy profile across layers, the authors test this probe's capability to discriminate among architectures, task types and text formats, through several experiments.

**Strengths:**

The paper is well written and the presentation is clear and well organized. The experimental settings are sound.

**Weaknesses:**

My main concerns with this paper are about novelty and usefulness. Regarding novelty, the idea of probing intermediate representations of transformers across layers has been used extensively in the literature. In particular, several papers, such as (Skean et al, "Does Representation Matter? Exploring Intermediate Layers in Large Language Models", 2024) and (Skean et al., "Layer by Layer: Uncovering Hidden Representations in Language Models", 2025) already use entropy as a probing measure, albeit using a different definition that directly involves hidden representations instead of first converting them to probability distributions using the final linear layer. (To be fair, I am not sure about which of the two approaches is preferrable, although it is not very clear why it is justified to apply the final linear layer, which has been trained to be at the end of the architecture, to intermediate representations.) The related mutual information measure has already been used in (Chang et al., "The Generalization Ridge: Information Flow in Natural Language Generation") as a measure of memorization in intermediate layers of transformers.

Regarding usefulness, my main concern is that, from the paper, it is not clear what is the point or advantage of introducing their entropy-based probe, compared to those already used in the literature. The paper mostly focuses on distinguishing architectures and tasks, which is not of fundamental importance, as those objectives can be achieved by much simpler means. Furthermore, this study does not shed light on the inner workings of transformers, apart from very generic remarks such as "We conjecture that high entropy phases, whether in the early or intermediate layers, allow the model to explore more possibilities in its response" at lines 292-293. It is also not clear why their choice of entropy is convenient compared to the other measures already used in the literature, as in the paper there is no comparison of performance between their probes and others.

**Questions:**

As expounded above, I would like the authors to address my concerns about the novelty and the usefulness of their method. In particular, I would like the authors to answer the following points:
1. How does your entropy probe compare to other measures already existing in the literature, such as the ones that I mentioned above? What is the advantage of your measure compared to them? Do you have any experiments showing the superiority of your probe?
2. What is the main purpose of using your probe? Can it helpful to interpret transformers' inner workings? Can it help improve future architectures in any way? Distinguishing architectures and tasks is interesting, but can it be useful in any practical, real-world setting?

---

> ### Author Response · Authors · 2025-12-03
>
> We thank the Reviewer for considering our presentation **excellent** and for recognizing the soundness of our experimental setup. Below we address the concerns regarding novelty, justification of our entropy computation, and usefulness.
>
> **Question 1**
> The prior works cited by the Reviewer probe hidden-state entropy directly in representation space. By contrast, our probe is, to the best of our knowledge, the first to define token-space entropy at every layer, by applying the model’s trained output head to intermediate representations. This yields probability distributions over the actual vocabulary, enabling a direct comparison across layers (hidden representations in different layers might have different meanings), models, tasks, and datasets in a common probabilistic space.
>
> This distinction is nontrivial: (1) Token-space entropy exposes model uncertainty in a semantically interpretable domain. Hidden-state entropy quantifies internal variability, but not how the model’s belief over tokens evolves. Token-space entropy instead exposes the richness or collapse of the model’s predictive distribution across layers. (2) Early-exit justification for using the final linear layer. Works on early-exit and intermediate decoding, like \[1\] that we cite in our paper, show that intermediate layers of LLMs naturally produce meaningful token logits when passed through the final LM head. Our approach is therefore consistent with the intrinsic structure of transformers and leverages a capability that has already been empirically validated.
>
> Regarding comparison to cited works, those probes were not developed for a common benchmark or task where superiority can be measured directly, and their objectives differ. Thus, a direct head-to-head comparison would be largely ill-posed. We agree that building a shared benchmark for probing comparison is an excellent direction for future work.
>
> \[1\] Shan, Weiqiao, et al. "Early exit is a natural capability in transformer-based models: An empirical study on early exit without joint optimization." *arXiv preprint arXiv:2412.01455*(2024).
>
> **Question 2**
> While prior work has considered related information-theoretic metrics, our goal is to examine more carefully what this particular form of token-space entropy captures and how far its interpretative power extends. Our results indicate several concrete ways in which the probe provides insight: (1) showing architecture-specific entropy signatures. Across all models we study, transformer families exhibit consistent and reproducible entropy profiles. To our knowledge, these family-specific shapes have not been documented before. We view this as evidence that entropy evolution reflects architectural inductive biases, which may inform future analyses of architectural design. Our focus here is not to claim immediate downstream performance benefits, but to clarify what information this metric carries. (2) Layerwise memorization behaviour. We use entropy to study how memorized content manifests across depth, providing a simple and interpretable signal of where memorized sequences influence the generation trajectory. (3) Relation to sampling dynamics. Token-space entropy is directly connected to the model’s predictive uncertainty. We show rises and drops in entropy. This yields practical diagnostic value: for example, highlighting layers where sampling paths become degenerate or overly deterministic. (4) Interpretability of intermediate computation. Entropy trajectories we observe align with meaningful shifts in the model’s processing (e.g., the transition from broad candidate sets to narrow commitments).
>
> Overall, our goal is not to propose a replacement for existing probes but to systematically characterize the behaviour and limits of this particular entropy measure. We believe that clarifying its properties provides value both for understanding existing applications (such as memorization across layers) and for motivating future ones. We agree that further development of these implications would strengthen the paper.
>
> In summary, our framework is novel in that it:
>
> - operates in **token space**
> - reveals **family-specific entropy patterns not previously observed,**
> - enable **per-layer memorization analysis**
> - provides practical insights into **uncertainty evolution** and **early-exit behavior**
>
> We thank the Reviewer again for the feedback.

---

### Meta-Review · Area_Chair_mNt9 · 2026-01-06

**Summary:**

The paper introduces an entropy-based method to obtain a curve (entropy for each layer) for an input text of an LLM, that can provide an insight into the inner workings of the computation. The authors investigate the connection between these curves and LLM families, task types, and output correctness.

The reviews mention the paper to be well written and easy to understand. They also appreciated thoroughness of the experiments in terms of the number of llms (e.g. r1uB “Several larger trained open source models are compared.”) and the clear and convincing setting.
The approach provided has mostly positive sentiment. It is mentioned to be convincing, e.g. r1uB “The entropy is indeed an interesting measure of the way information is processed in the network”, PJ6w “The central idea (using the "entropy profile" to identify attributes of models/use-cases) is novel and well-motivated.”, rW4J  “The approach is straightforward and transparent, and grounded at a basic level in information theory”. Reviewer CJe4 in contrast raised some concerns regarding the approach, in particular its novelty when compared with recent existing publications. The authors provided an explanation in the rebuttal which seems to address the issue, explaining the inherent difference between their approach, that captures the computation, to other approaches, that capture the input. Given the positive sentiment from other reviews I see this issue of novelty as minor, since it is inherently subjective. However, I urge the authors to properly address all works mentioned in the review in their paper, and thoroughly explain the differences there.

The main concern, raised in some way by all four reviewers is the overall impact / contribution. My summary of this weakness is that the paper provides interesting correlations related to the proposed approach, but fails to give a sufficiently actionable insight or concrete conclusion. In CJe4’s review this issue is not directly addressed, but the request to better understand the advantage of this approach over others does reflect it. rW4J mentions  “While the results are convincing and the methodology solid, what is less developed in the paper are the implications and/or applications.” PJ6w “... it is unclear what the higher goal here is, or how it fits towards the broader goal of understanding the inner workings of transformers. They identify and observe interesting patterns … offer little to no insights as to how/why these patterns show up.” r1uB “To me, a successful interpretable analysis would connect instance-based observations and phenomena to the type of the curve, and validate that the entropy curve has some predictive properties. ”

This major concern appears to be the reason for the overall low scores, and the authors did not provide convincing counter arguments to this weakness in the rebuttal. Given this, I think the paper has potential given its mentioned strengths, but requires a more coherent analysis before it reaches the publication bar for ICLR.

**Reviewer Concerns:**

As stated above, the major concern, the need for more refined insights, remain.

**Reviewer Scores:**

Since the major concern, raised in all reviews remained, I don't believe the scores would have changed

---

### Decision · Program_Chairs · 2026-01-26

Reject